# A Set of Dysregulated Target Genes to Reduce Neuroinflammation at Molecular Level

**DOI:** 10.3390/ijms23137175

**Published:** 2022-06-28

**Authors:** Marcella Massimini, Benedetta Bachetti, Elena Dalle Vedove, Alessia Benvenga, Francesco Di Pierro, Nicola Bernabò

**Affiliations:** 1Faculty of Veterinary Medicine, University of Teramo, 64100 Teramo, Italy; 2R&D Division, C.I.A.M. Srl, 63100 Ascoli Piceno, Italy; benedetta.bachetti@nilitaly.com (B.B.); elena.dallevedove@nilitaly.com (E.D.V.); alessia.benvenga@nilitaly.com (A.B.); 3Velleja Research, 20125 Milan, Italy; f.dipierro@vellejaresearch.com; 4Digestive Endoscopy Unit and Gastroenterology, Fondazione Poliambulanza, 25124 Brescia, Italy; 5Faculty of Bioscience and Technology for Food, Agriculture and Environment, University of Teramo, 64100 Teramo, Italy; nbernabo@unite.it

**Keywords:** preclinical screening, in vitro alternative methods, neurodegeneration, transwell co-culture, inflammation

## Abstract

Increasing evidence links chronic neurodegenerative diseases with neuroinflammation; it is known that neuroprotective agents are capable of modulating the inflammatory processes, that occur with the onset of neurodegeneration pathologies. Here, with the intention of providing a means for active compounds’ screening, a dysregulation of neuronal inflammatory marker genes was induced and subjected to neuroprotective active principles, with the aim of selecting a set of inflammatory marker genes linked to neurodegenerative diseases. Considering the important role of microglia in neurodegeneration, a murine co-culture of hippocampal cells and inflamed microglia cells was set up. The evaluation of differentially expressed genes and subsequent in silico analysis showed the main dysregulated genes in both cells and the principal inflammatory processes involved in the model. Among the identified genes, a well-defined set was chosen, selecting those in which a role in human neurodegenerative progression in vivo was already defined in literature, matched with the rate of prediction derived from the Principal Component Analysis (PCA) of in vitro treatment-affected genes variation. The obtained panel of dysregulated target genes, including *Cxcl9 (Chemokine (C-X-C motif) ligand 9)*, *C4b (Complement Component 4B)*, *Stc1 (Stanniocalcin 1)*, *Abcb1a (ATP Binding Cassette Subfamily B Member 1)*, *Hp (Haptoglobin)* and *Adm (Adrenomedullin)*, can be considered an in vitro tool to select old and new active compounds directed to neuroinflammation.

## 1. Introduction

Neurodegenerative diseases represent a public health and socio-economical issue; progressive degenerative processes have a major impact on the lives of many people, both personally and professionally, leading to a total inability to exercise everyday activities. Nowadays, as a result of increased life expectancy and changed population demographics, neurodegenerative diseases are becoming more common, affecting millions of people worldwide [1]. Alzheimer’s disease (AD) and Parkinson’s disease (PD) are the most common neurodegenerative disorders and account for a significant and increasing proportion of morbidity and mortality, as well as representing a high economic burden [2]. The challenge lies in the development and implementation of neuroprotective therapies that can slow disease progression. Moreover, the discovery of early biomarkers linked with neuroinflammatory diseases would help diagnosis and improve treatment response rates.

The management of clinical symptoms through drugs, diet, and supplements decreases the progression of cognitive disorders by various mechanisms, including a reduction in oxidative stress and inflammation, or improved mitochondrial and neuronal function [3,4].

A multidisciplinary approach between in vitro, in vivo, and ex vivo studies may bridge the gap between experimental models and the complexity of disease; in vitro models could provide a reliable approach for understanding the mechanisms, and testing treatments, of neurodegenerative diseases [5]. In this context, the murine neurodegeneration model remains the most used to date, along with the latest in vitro organotypic or multicellular culture models that better resemble the in vivo situation [6,7,8]. In particular, the co-culture systems allow an understanding of the effects induced by activated microglia soluble factors on the neuronal integrity and function, in normal and dysregulated conditions [9]. Synaptic loss due to microglia inflammation is strongly correlated with cognitive decline in different mouse models of AD [10]; inflamed microglia cells in a triculture system directly stimulate axons phagocytosis and synaptic loss, determining the death of 50% of the neurons and astrocytes in the close proximity [11]. The resident microglia of the mammalian central nervous system are involved in both neuroprotection and pathological conditions, but extensive data show that microglial hyperactivation can promote tissue damage [12].

The aim of this work is to provide an in vitro model to screen neuroprotective agents, selecting a group of inflammatory marker genes linked to neurodegenerative disorders using next generation sequencing (NGS). More specifically, HT22 hippocampal cells were stimulated with LPS/IFNγ inflamed microglia and the main upregulated and downregulated genes in both cells were identified through Lexogen Differentially Expressed Genes (DEG) analysis. Among them, a set of inflammatory marker genes was identified through a deep search of their functions in the literature, selecting those in which a contribution to neurodegenerative progression was already defined, together with their role as potential target therapeutic genes after α-lipoic acid, propentofylline, and N-acetyl-cysteine treatment evaluation [13,14,15].

## 2. Results

### 2.1. Hippocampal Differentiation

The neuron cell differentiation was evaluated through *Grin1 (Glutamate ionotropic receptor NMDA type subunit 1)* gene expression. The *Grin1* mRNA levels were significantly increased in the differentiated hippocampal HT22, compared with the control cells (1.344 ± 0.48 compared with 0.928 ± 0.15 *p* = 0.045 unpaired *t*-test) as shown in Figure 1.

### 2.2. Effect of Stimulated BV2 Cells on Differentiated HT22 Cell Viability

A Kruskal–Wallis test showed the effect of the conditioned medium, obtained from inflamed BV2 microglia, on HT22 cell viability. The stimulus affected the HT22 cell viability (*p* < 0.001) with a significant decrease, shown by a post hoc comparison test after 24 h of conditioned medium exposure (67.57 ± 2.53 *p* < 0.01), as shown in Figure 2.

### 2.3. Co-Culture Principal Differentially Expressed Genes and In Silico Evaluation

The HT22 neuronal cells and inflamed BV2 microglia cells were co-cultured and, with the aim of obtaining a panel of dysregulated inflammatory genes, modulation of the principal DEG was identified through Lexogen mRNA sequencing in both of the cell lines (Appendix A). The full gene names will not be always reported in the manuscript, considering that the Ensembl identification number (ID) is present in Appendix A.

The modulation of DEG, identified through mRNA sequencing, shown in Appendix A, was confirmed by repeating the co-culture experiment and comparing the gene expression with non-inflamed cells cultured in a standard monolayer. More specifically, with the aim of understanding the effect of co-culture on gene expression, both HT22 and BV2 cultured in standard monolayer were stimulated with LPS 100 ng/mL and IFNγ 5 ng/mL for 24 h. Appendix A shows the effect of the “inflammation” and “co-culture” factors and the interaction between them. The interaction effects represent the combined effects of factors on the outcome, in our case the two factors are “inflammation” and “co-culture”. It is important to take into account this effect, since the interaction can affect the dependent variable.

The significant differences that emerged among the group, through post hoc testing, are shown in Appendix A, that shows the fold change ± SEM of the inflammatory marker genes in HT22 and BV2 cells compared with the non-inflamed cells cultured in standard monolayer.

For each set of genes, the corresponding proteins were identified. By using a specific bioinformatic tool (String), we built up the networks representing the interactions among these proteins (Figure 3, left side), and among the proteins that are interacting with them (after four enrichment cycles) (Figure 3, right side). The enrichment showed us the main biological processes where the our proteins of interest are key players, such as immune response, cell communication, and adhesion.

Next, we compared the set of up- and downregulated genes in the two different cellular models (Figure 4), finding that one gene is downregulated in both systems, while eight are upregulated. Finally, we found the proteins corresponding to those genes and studied their interaction before and after four enrichment cycles, observing their involvement in the cytokine-mediated response process (Figure 5).

### 2.4. Effect of Bio-Actives on Cells Viability

To evaluate the potential role of the abovementioned genes as target therapeutic genes, the effects of α-lipoic acid, N-acetyl-cysteine, and propentofylline on HT22 and BV2 cell viability were tested (Figure 6). These bio-actives were chosen for their ability to reduce neuroinflammation [16,17,18]. The cells were exposed for 24 h to the following treatments: α-lipoic acid at 0.1 mM, 0.5 mM, 1 mM, 2 mM; N-acetyl-cysteine at 0.1 mM, 0.5 mM, 1 mM, 2 mM; propentofylline at 0.1 mM, 1 mM, 10 mM, 100 mM. A Kruskal–Wallis test showed that the exposure to the α-lipoic acid induced a significant change in the percentage HT22 (*p* < 0.01) and BV2 (*p* < 0.01). Post hoc comparisons using the Bonferroni test revealed a significant decrease in the groups treated with α-lipoic acid at 2 mM dosage in HT22 (43.79 ± 9.08, *p* < 0.01), at 1 mM (58.20 ± 8.16 X, *p* < 0.05), and at 2 mM (52.25 ± 1.83, *p* < 0.01) in BV2 cells. N-acetyl-cysteine significantly affected the HT22 and BV cell viability (both with *p* < 0.001). The Bonferroni comparison test revealed an increase in cell viability with respect to the control at 0.1 mM dosage exposure in HT22 (112.80 ± 4.06, *p* < 0.05) and at 2 mM (138.65 ± 18.89, *p* < 0.001) in HT22, and a decrease at 2 mM dosage (129.92 ± 4.67, *p* < 0.05). Finally, a significant effect in the percentage viability was observed after propentofylline treatment in both of the cell lines (HT22 *p* < 0.001; BV2 *p* < 0.05). A post hoc comparison test revealed a decrease in cell viability with respect to the control at the higher dosages in both of the cell lines (in HT22 at 100 mM (76.58 ± 7.64, *p* < 0.05) and in BV2 at 100 mM (69.11 ± 6.56, *p* < 0.05)).

### 2.5. Effect of Bio-Actives on Gene Expression and PCA Analysis

The effects of α-lipoic acid, N-acetyl-cysteine and propentofylline on HT22 and BV2 cell viability were tested at the chosen dosages for their ability to reverse the dysregulation induced by inflammation in a co-culture of the identified genes. The effects of active principles treatment on gene expression in inflamed co-culture cells are shown in Table 1 and Table 2.

The main goal was to check whether some of the genes vary together. This would reduce the number of variables in our model. In our dataset, we have the genes as variables and the treatments as samples. To reduce the number of variables, while increasing the interpretability of results, the PCA is used in this work. By the means of this technique, new uncorrelated variables are created, which maximize the variance while minimizing potential information loss. The results showed that more than 70% of the variability is explained with only two dimensions. More specifically, we have 70.71% for the HT22 cell line and 71.95% for the BV2 cell line. Looking at the correlation circle and the correlation plot for HT22, it is evident that some genes vary together; we found that *Cxcl9*, *Iigp1*, *Gbp5*, *Gbp11*, *Ifi44*, *Igtp*, *Usp18*, *Cxcl11*, *Ifit47*, *Oas2,* and *Gbp8* have a high impact on the first principal component. All of these genes have a good quality of representation in the new coordinate system, they are close to each other and overlap. We can conclude, therefore, that evaluating one of these genes could predict the trend of the others. The genes *Mlycd*, *Gpr56*, *Selenbp1*, *Serpine1*, *Stc1,* and *Milr1* have a good quality of representation in the first two principal components; *Mlycd* could also be evaluated to predict the trend of *Hp* and *Gpr35*, which have a lower quality of representation but vary together with *Mlycd*. Looking at the graph of individuals, there is a clear distinction between the inflamed and the non-inflamed samples, regardless of the treatment. We can also discriminate clusters, as shown in Figure 7.

Regarding the BV2 PCA graph of variables, we appreciate that a large number of the genes are highly correlated, so we can assume that they vary together. The behavior of the genes *Ly6a*, *Gbp8*, *Cxcl9*, *Serpine3g*, *Ly6c*, *Gbp4*, *Gbp2*, *Cxcl11*, *Slamf8*, and *Gbp5* can be predicted by looking at the behavior of one of them. Likewise, the genes *Cxcl17*, *Spns2*, *C5ar1*, *Slap2*, *Gpr97*, *Cxcr2*, *Pad2,* and *Lix1* are correlated. The genes *Adm*, *Kif17*, *Gpr183,* and *Myo7c* have a good quality of representation, but they are not correlated as much as the two other groups of genes mentioned above. Regarding the graph of individuals, three comparable clusters can be identified in the HT22 graph: at center right are the inflamed samples treated with the active principles, N-acetyl-cysteine and propentofylline; on the bottom right is a cluster with control samples (both inflamed and non-inflamed), non-inflamed treated samples, and inflamed α-lipoic acid-treated samples; on the top left is the last cluster with the non-inflamed α-lipoic acid-treated samples (Figure 8).

## 3. Discussion

The aim of the study was to select a set of inflammatory marker genes linked to neurodegenerative diseases with the intention of providing a means for the preclinical screening of active compounds. For this purpose, differentiated mouse hippocampal neuronal cells were stimulated with LPS/IFNγ inflamed microglia and the top twenty differentially expressed genes in both cells were identified through Lexogen mRNA sequencing. The main inflammatory processes were shown through subsequent in silico analysis, including leucocytes’ chemotaxis, defense responses, and regulation of calcium homeostasis. A selection of marker genes was obtained by exposing the inflamed co-culture to the effect of three active principles known for their ability to reduce neuronal inflammatory processes [13,14,15]. The total number of genes was reduced, selecting those that had a high impact in the model measured through PCA, and that were also identified in previous literature as playing a role in neurodegenerative progression. This panel of genes is represented in Table 3.

Some of the differentially expressed genes were found to be dysregulated in both of the cell lines. Two of them are *Cxcl9* and *Cxcl11*, which are involved in Th1-type response, correlating with T-cell infiltration; the comprehension of CXCR3 (C-X-C Motif Chemokine Receptor 3)/CXCL9′s binding mechanism of action is considered crucial for the treatment of central nervous system diseases [41]. It has been observed that CXCR3 upregulation promotes plaque formation and behavioral deficits in an AD model [20]. The literature also shows that CXCL9 and CXCL10 might be involved in a neuronal–glial interaction, considering that they are upregulated in human AD brains [19,21]. The *Iigp1* and *Gbp5* genes, both upregulated in both of the cell lines of our model, are linked to neurodegeneration; *Iigp1* deregulation has been found in the 5XFAD early onset mouse model and *Gbp5* is upregulated in mice microglia after Aβ peptide exposure [22,23,24]. Microglia shares phenotypic characteristics with peripheral monocytes and it exists in various states of activation; in fact, resident microglia or macrophages infiltrating from the circulation become polarized towards a pro-inflammatory (M1) phenotype upon exposure to the pro-inflammatory cytokines IFN-γ, TNF-α, and cellular or bacterial products, such as LPS [42]. In the work of Orecchione et al., in vitro and in vivo M1 macrophage gene expressions were compared. Among the shared upregulated genes, there were many IFN γ-induced genes—including *Cxcl9—*highlighting its strong and important involvement in vivo in the immune response processes [43]. The other genes that have been upregulated in both cell lines, such as *Gbp8*, *GM4951,* and *Serpine3G,* belong to the same molecular pathway of the above-mentioned genes. To the best of our knowledge, they have not been found directly involved with or correlated with neurodegeneration. However, we observed that *Gbp8*, together with *Cxcl9*, *Cxcl11*, *Iigp,* and *Gbp5* have been reverted by treatment. Furthermore, the *Susd1* gene was downregulated after inflammatory stimulus in both of the cell lines, but no relationship with neurodegenerative disorders is present in the literature.

Among the genes that have been modulated in the HT22 cell line only, results showed the following to be correlated with neurodegeneration: *Ifi44; C4b*; *Stc1*; *Nlrc3*; *Abcba1*; *Hp;* and *Gpr56*. The region around the *Ifi44* gene showed open chromatin architecture in an in vivo mouse model of AD, supporting the upregulation observed in our model [25,26]. Regarding *Complement C4B*, an increased expression of the fourth serum complement component 4 (C4) has been observed in AD patients in many studies; the two isoforms of the proteins are encoded by two genes, localized to the HLA class III region, that both show copy number variations (CNVs), influencing the C4 protein levels [27]. Stanniocalcin 1 is a nerve cell-enriched protein involved in intracellular calcium homeostasis regulation; changes in calcium regulation are hypothesized to play a role in the pathophysiology of AD; cerebrospinal fluid STC-1 showed an increasing trend in AD patients and, in vitro, it is αβ-induced [28,29]. Moreover, the *Nlrc3* gene was strongly downregulated in our system; a temperature shift assay reveals that the late onset of *Nlrc3*-like deficiency leads to excessive microglia cell death, which is attributed to the aberrant activation of a canonical inflammasome pathway, indicating that the proper regulation of the inflammasome cascade is critical for the maintenance of microglia homeostasis [30]. The ATP-binding cassette (ABC) transporters, in particular *p*-glycoprotein (encoded by *Abcb1*), are important and selective elements of the blood–brain barrier (BBB), and actively contribute to brain homeostasis. Over the last decade, a number of reports have shown that *Abcb1* actively mediates the transport of αβ-peptide [32]. Cirrito et al. demonstrated that the deficiency of *Abcb1* at the BBB increased αβ deposition in an AD mouse model, suggesting that αβ is transported out of the brain or periarterial interstitial fluid through this transport system [31]. Haptoglobin, encoded by the *Hp* gene, is an acute-phase protein of inflammation that binds ApoE and influences the key roles of this apolipoprotein in cholesterol homeostasis. Like other extracellular chaperones, haptoglobin colocalizes with amyloid plaques and inhibits Aβ fibril formation in vitro [33]. Regarding the BV2 cell line, our results showed that several of the inflammation-modulated genes of our set are involved in neurodegeneration: *Ly6a*, *Cxcl9*, *Serpine3G*, *Fam26f*, *Gbp2*, *Slamf8*, and *Adm.* The *Ly6a*-encoded protein is an acetylcholine receptor binding, and it was upregulated in a mouse model of sleep deprivation and AD [34]. The serine protease inhibitor A3G is overexpressed in the hippocampus of a mouse model of AD (5×FAD) and can be downregulated with a vitamin D supplement [22,36]. *Fam26f* or *Calhm6* (calcium homeostasis modulator family member 6) is overexpressed in mouse microglia in response to Th1-derived factors [37]. *Slamf8,* involved in the toxic inflammatory response induced by the Aβ peptide that accelerates synaptic and neuronal injury, together with the previously mentioned *Gbp5* gene, is upregulated in mice microglia after Aβ peptide exposure [24]. Meanwhile *Gbp2* upregulation is associated with neuron apoptosis in the rat brain cortex following traumatic brain injury [38]. Adrenomedullin is downregulated in our system, but its role in neurodegeneration is still controversial. It is a potent vasodilator peptide, and it is considered to be a neuromodulator with antiapoptotic and antioxidant properties that can protect the brain from damage. The *Adm* gene’s upregulation exerts a neuroprotective action in the brain, mediated by the regulation of nitric oxide synthases, matrix metalloproteases, and inflammatory mediators [39]. In contrast, it has been observed in AD patients, that neural cytoskeleton failure is associated with an increase of adrenomedullin levels, resulting in axon transport collapse and synaptic loss [40].

In our opinion, due to the validation completed by using putative actives like lipoic acid, N-acetyl-cysteine, and propentofylline, already described as exerting a protective role against neuronal chronic degeneration, this co-culture model has the advantage of providing a suitable in vitro tool to select molecules that are potentially candidates to act against neuroinflammation. It is noteworthy, one of the most important benefits of our model is the rate of prediction that can be derived by the PCA analysis of genes’ variation. Most of the genes vary with an apparently clear correlation, allowing the researchers to reduce the number of genes to be checked.

The possible limitations of our in vitro model are concerned with not taking the astrocytes into proper consideration, the nervous system cells known to affect neuroinflammation process and response, and to have selected the importance of some genes also in direct relationship with the results obtained by using lipoic acid, N-acetyl-cysteine, and propentofylline, assuming that these are absorbed unchanged, and in that unchanged state then reach the central nervous system. Indeed, it cannot be excluded that their metabolites, instead of the native molecules, are the real effectors in central nervous system, and that the genes that are involved could be different.

## 4. Materials and Methods

### 4.1. Cell Lines Maintenance and Hippocampal Differentiation

The immortalized murine HT22 neuron cells were cultured in Dulbecco’s modified Eagle’s medium (Corning, New York, NY, USA) supplemented with 10% fetal bovine serum (FBS) (Corning), 100 U/mL penicillin (Corning), 0.1 mg/mL streptomycin (Corning). The hippocampal differentiation was performed in modified, serum-free medium (Dulbecco’s modified Eagle’s medium, 1× N2 supplement (Life Technologies AG, Carlsbad, CA, USA)), 50 ng/mL nerve growth factor-β (NGF-β), 100 μM phorbol 12,13-dibutyrate (PDBu; Santa Cruz Biotechnology Inc., Dallas, CA, USA), 100 μM dibutyryl cyclic adenosine monophosphate (cAMP) (Santa Cruz Biotechnology Inc.), 100 U/mL penicillin, as well as 0.1 mg/mL streptomycin (Corning). All of the treatments were performed in pre-warmed (37 °C) differentiation medium with reduced NGF-β content (5 ng/mL) [44,45]. The differentiation was evaluated through Nr1 (N-methyl-D-aspartate receptor 1) gene expression, as shown in Section 2.5. The murine BV2 microglia cells were obtained from the Interlab Cell Line Collection, Banca Biologicae Cell Factory, Italy, and cultured using Roswell Park Memorial Institute medium 1640 (RPMI) supplemented with 10% FBS (Sigma, St. Louis, MO, USA), 2 mM L-glutamine (Sigma), 100 mM sodium pyruvate (Sigma), 100 U/mL penicillin, and 100 mg/mL streptomycin (Sigma) in a 5% CO_2_ incubator at 37 °C.

### 4.2. Inflammatory Stimuli Effects on Cell Viability

The effect of the conditioned medium obtained from the inflamed microglia on the HT22 cell viability was measured using the MTT assay. The BV2 cells were seeded at a density of 2000 cells/well in 96-well cell culture plates and, after 24 h, were stimulated with LPS (lipopolysaccharide) 100 ng/mL and IFNγ 5 ng/mL for 24 h; the HT22 hippocampal neurons, seeded at a density of 5000 cells/well in 96-well cell culture plates in differentiation medium for 24 h, were exposed to BV2-inflamed supernatants for 3, 6, 24, and 48 h [46]. At the end of each incubation, the supernatants were replaced with 0.1 mL of fresh medium without phenol red, containing 0.5 mg/mL of MTT (3-(4,5-Dimethylthiazol-2-yl)-2,5-Diphenyltetrazolium Bromide); the plates were then returned to the incubator for 4 h and were gently shaken occasionally. The crystals of formazan (the MTT metabolic product) were solubilized by 0.1 mL ethanol/DMSO 1:1 lysis buffer and spectrophotometrically quantified at a wavelength of 570 nm with the reference at a wavelength of 695 nm. The differences in cell growth were measured as a percentage of growth rates of stimulated cells compared to unstimulated cultures.

### 4.3. HT22/BV2 Co-Culture Inflammation and mRNA Sequencing

The HT22 and BV2 cells were cultured in two chambers separated by a 0.4 μm semi-permeable membrane allowing diffusion of the soluble factors between the compartments (trans-well system Greiner). On the first day, 200,000 HT22/well in differentiation medium and 80,000 BV2/well in complete growth medium were seeded, respectively, in the wells and in the upper chamber. On the second day, the BV2 cells were inflamed with LPS 100 ng/mL and IFNγ 5 ng/mL. On the third day, the HT22 differentiation medium was replaced with the same differentiation medium with reduced NGF-β content (5 ng/mL); the upper chambers were then moved in the HT22 wells and cultured together with BV2 for 24 h. The control group was cultured and moved in the same way without the inflammation step. Lexogen utility was chosen for mRNA sequencing analysis. To confirm the results, expression of modulated genes identified through Lexogen mRNA sequencing was evaluated with Real-Time PCR, as described in the following Section, by repeating the co-culture experiment and adding two control conditions in addition to non-stimulated co-culture. More specifically, with the aim of understanding the effect of co-culture on gene expression, both HT22 and BV2, cultured in standard monolayer culture, were stimulated with LPS 100 ng/mL and IFNγ 5 ng/mL for 24 h.

### 4.4. Pellet, mRNA Isolation and Gene Expression

The total RNA was isolated using the Total RNA Purification Kit (Norgen, Thorold, ON, Canada), following the manufacturer’s recommended protocol. Starting with 1 μg of RNA templates, first strand complementary DNA (cDNA) was synthesized using SensiFAST cDNA Synthesis Kit (Abm Good). The relative abundance of the *Nr1* gene was evaluated by Real Time PCR using the PowerUp™ SYBR™ Green Master Mix on a QuantStudio5 instrument (Applied Biosystem, Waltham, WA, USA) in both cell lines. The primers used for the amplification are reported in Appendix A and all of the data were normalized to the endogenous reference gene *Gapdh* [47,48,49,50,51,52,53,54,55,56,57,58,59,60,61,62,63,64]. A May-Grünwald-Giemsa stain was also performed for the observation of neurite growth Briefly, the cells were seeded on a sterilized coverslip in a 15 mm cell culture dish, differentiated as explained in Section 2.1, and then fixed with methanol. The cells on the coverslip were stained with five subsequent immersions in the May-Grünwald and five subsequent immersions in the Giemsa. After being dried, the coverslip was mounted on a slide with Biomount.

### 4.5. Network Creation and Enrichment Analysis

To explore the interaction among the proteins from the up- and downregulated genes, and to identify and predict the new molecules possibly involved in the proteasome network, here we used a bioinformatic approach based on the network’s theory. In brief, a network is a set of nodes—the proteins—linked by edges—the protein–protein interactions. It gives rise to a mathematical model that could be studied by computing its topological properties, i.e., the statistical descriptor of the parameters related with the network architecture both at a global and local level. To aim as the data source, we used the Search Tool for the Retrieval of Interacting Genes/Proteins (STRING) [65]. STRING is a database of known and predicted protein interactions. They include direct (physical) or indirect (functional) associations, and are derived from different sources: genomic context, high-throughput experiments, conserved co-expression, and previous knowledge. First, we obtained the networks representing the interactions among the proteins, filtering the information for Mus musculus (house mouse), with a medium confidence score (0.400). Once the networks were obtained, we carried out four cycles of enrichment, setting the inflation parameter at four. Thus, we expanded the network by also including new proteins, based on the molecular data provided by STRING. Finally, we made a further analysis aimed at identifying the clusters of molecules interacting within the network, using a Markov Cluster Algorithm (MCL), setting the inflation parameter at four. Once the network analysis was completed, we obtained the list of proteins taking part in the biological event we were studying and we carried out a functional enrichment analysis, using the following archives: Biological process, Molecular Function, and Cellular Component (all from Gene Ontology). This procedure enabled us to obtain the list of biological processes, molecular functions, and cellular components where the proteins of our interest were key players.

### 4.6. Bioactives Dosage Information and Effects on Cell Viability

HT22 and BV2 were treated with bioactive molecules α-lipoic acid, N-acetyl-cysteine and propentofylline, already known for their neuronal activity. α-lipoic acid, dissolved in ethanol, and N-acetyl-cysteine and propentofylline, dissolved in distilled water, were added at the following concentrations: α-lipoic acid 0.1 mM, 0.5 mM, 1 mM, 2 mM; N-acetyl-cysteine 0.1 mM, 0.5 mM, 1 mM, 2 mM; propentofylline 0.1 mM, 0.5 mM, 10 mM, 100 mM for 24 h. The culture medium, containing the vehicles alone, was added to the controls under the same conditions. To determine the mixture effect on cell proliferation, 7500 cells/well were seeded onto 96-well plates. At the beginning of the experiment, complete growth medium was replaced with fresh medium, containing the bio-actives or vehicles. After 24 h of incubation, the supernatants were replaced with 0.1 mL of fresh medium without phenol red, containing 0.5 mg/mL of MTT; the plates were then returned to the incubator for four hours and were gently shaken occasionally. The crystals of formazan (the MTT metabolic product) were solubilized by 0.1 mL ethanol/DMSO 1:1 lysis buffer and spectrophotometrically quantified at a wavelength of 570 nm with the reference at a wavelength of 695 nm. The differences in cell growth were measured as a percentage of growth rates of treated cells compared to untreated cultures.

### 4.7. Co-Culture Treatments with Bio-Active Molecules

HT22 and BV2 co-cultures were repeated as shown in Section 2.4, adding the bioactive molecules at the following concentrations: α-lipoic acid 0.5 mM; N-acetyl-cysteine 2 mM; propentofylline 0.001 mM. The treatments were added directly to the BV2 medium together with the inflammatory stimuli (day 2; see Section 2.3)

### 4.8. Statistical Analysis

To establish how the co-culture of BV2 and HT22 compared with standard monolayer culture as a representation of neuronal inflammation, a non-parametric two-way analysis of variance (ANOVA) was performed for each gene of the set. The factors considered were “inflammation” (two levels: yes and no) and “culture type” (two levels: standard and co-culture). When a significant ANOVA was obtained, a post hoc test was performed to check the differences among groups. If one interaction term was not significant, a one-way ANOVA was run on the other significant term, followed by a paired t-test when a significant *p*-value was found. We performed a Kruskal–Wallis test to select the adequate dosage of treatments among the concentrations used in the MTT assay. When the *p*-value was significant, an unpaired *t*-test was used to enlighten the differences among concentrations. The treatments on co-cultures were analyzed with Kruskal–Wallis test; Dunn’s test revealed the differences among treatments when a significant *p*-value was obtained. Due to the large number of genes involved in the model, a PCA analysis was run (BV2 and HT22) to reduce the high number of variables, compared to treatments. All of the statistical analysis were performed through R software (R Core Team (2021)).

## 5. Conclusions

We have described a dual culture model, using hippocampal neurons and microglia. Its response to co-culture, LPS, and INF-γ exposure, and to testing with putative anti-inflammatory molecules, provided a panel of genes which modulation could be considered predictive for neuroprotective drugs’ selection. The model could be used to select old and new drugs before testing them in animal models of neuroinflammation.

## Figures and Tables

**Figure 1 ijms-23-07175-f001:**
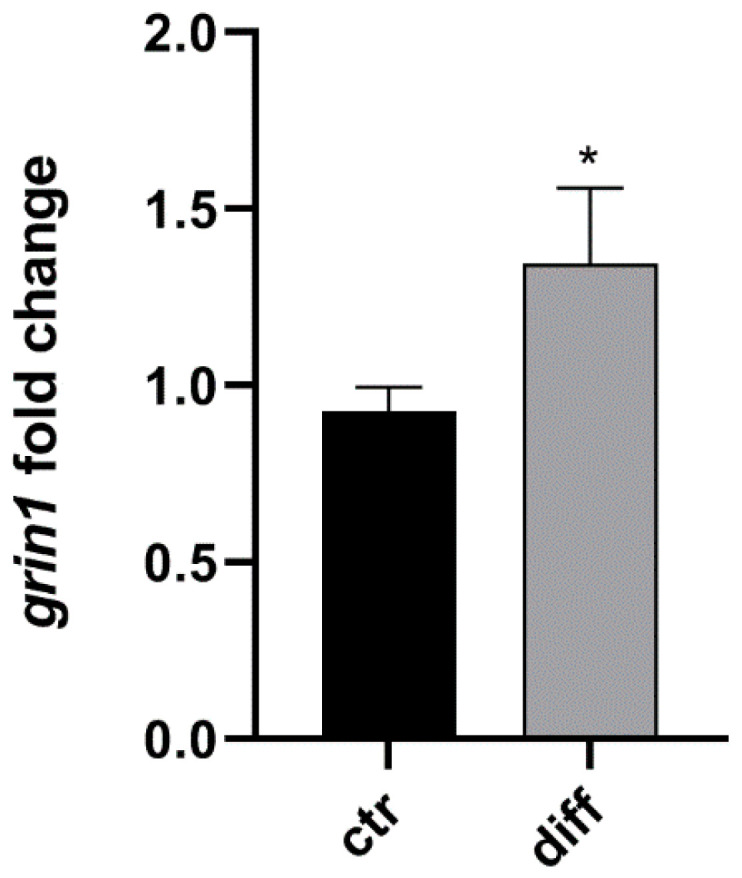
*Grin1* gene expression in differentiated hippocampal neuron. The bar graphs show the mean + SEM of three independent experiments performed in triplicate. The statistically significant differences between experimental conditions and untreated control cells are shown by asterisks (* *p* < 0.01).

**Figure 2 ijms-23-07175-f002:**
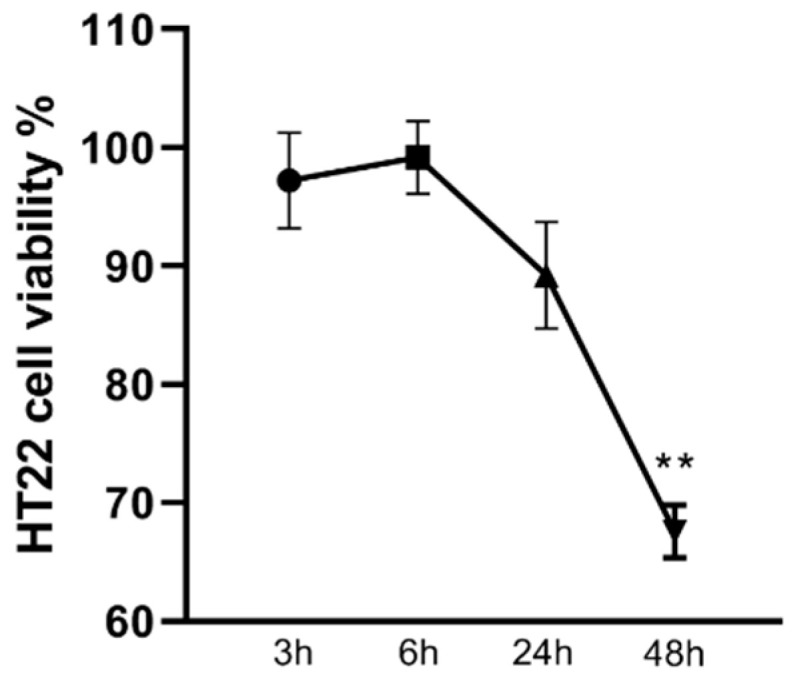
The effects of conditioned medium obtained from BV2 inflamed microglia on HT22 cell viability. Cell viability is expressed as the percentage of untreated cells (vehicle alone = 100% cell viability). Graph shows the mean + SEM of three independent experiments performed in triplicate. The statistically significant differences between experimental conditions and untreated control cells are shown by asterisks (** *p* < 0.01).

**Figure 3 ijms-23-07175-f003:**
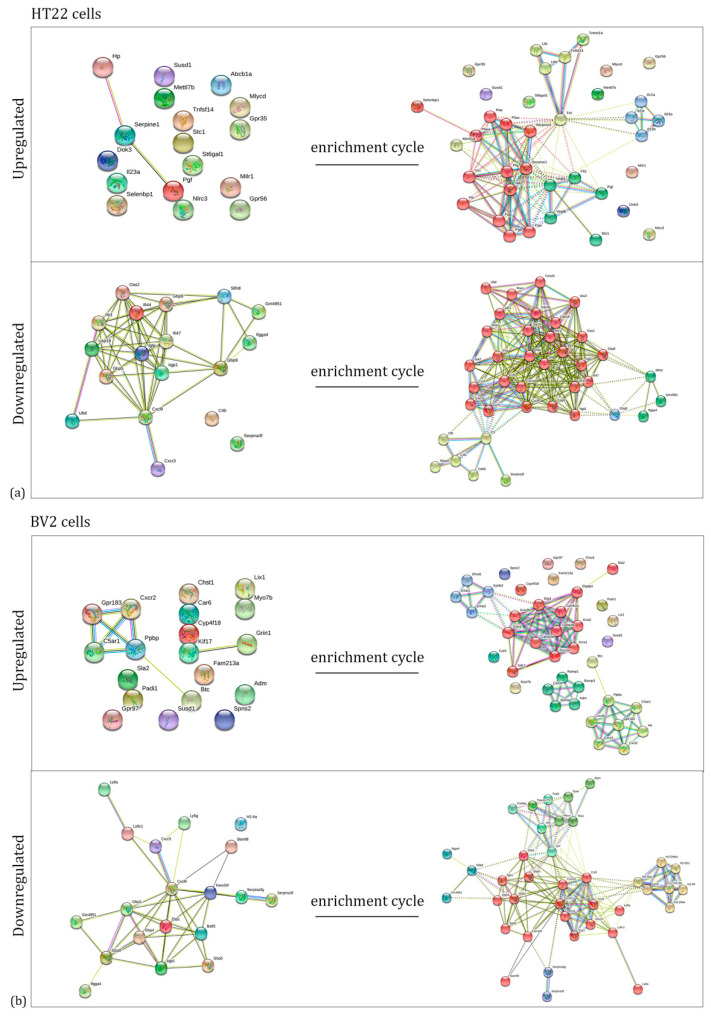
Networks representing the interactions among the proteins corresponding to the selected genes before (left) and after the enrichment procedure (right) in HT22 (**a**) and BV2 (**b**) cells.

**Figure 4 ijms-23-07175-f004:**
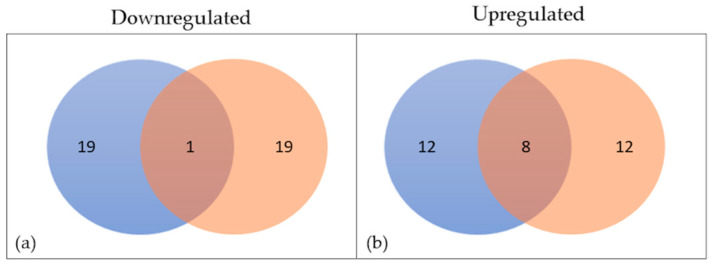
Venn diagrams showing the number of genes downregulated (**a**) or upregulated (**b**) in the two cellular systems; HT22 cells in blue and BV2 in orange.

**Figure 5 ijms-23-07175-f005:**
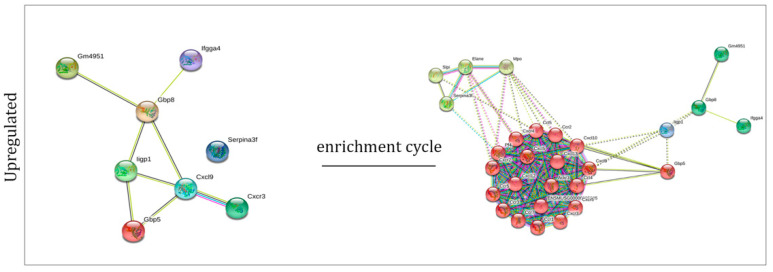
Networks representing the interactions among the proteins corresponding to the selected genes before (left) and after the enrichment procedure (right).

**Figure 6 ijms-23-07175-f006:**
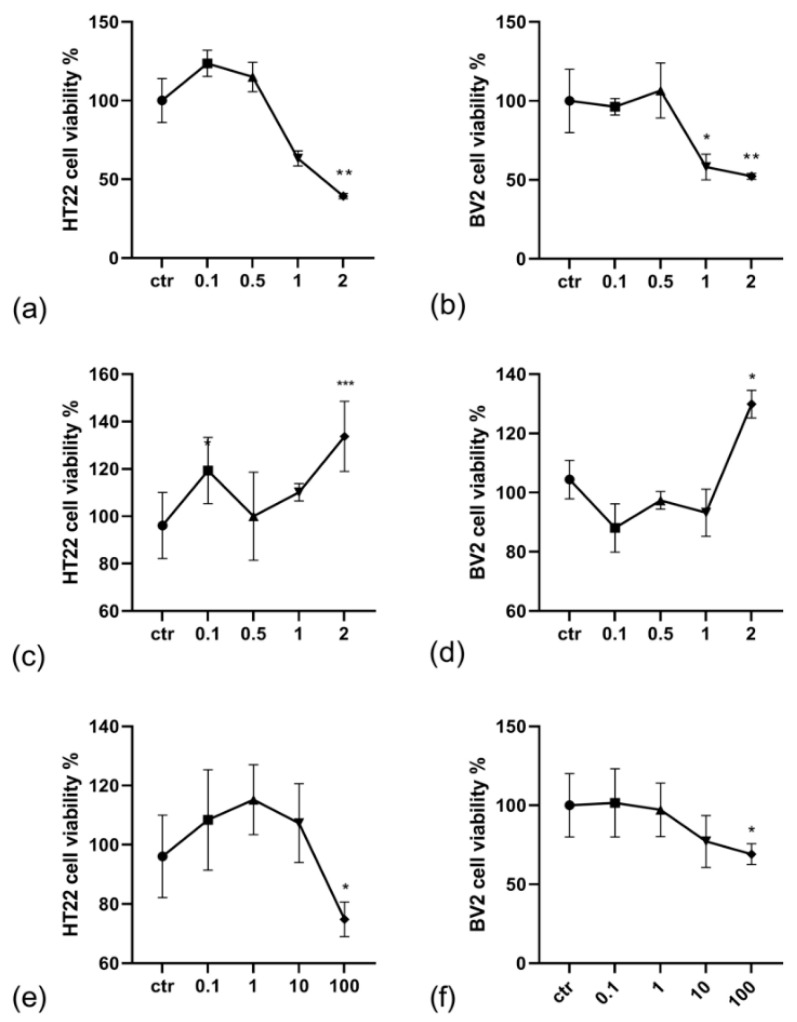
The effects of active principles on viability of HT22 and BV2 cells. HT22 (**a**,**c**,**e**) and HT22 and BV2 cells (**b**,**d**,**f**) were treated with increasing concentrations of α-lipoic acid (**a**,**b**: 0.1, 0.5, 1, 2), N-acetyl-cysteine (**c**,**d**: 0.1, 0.5, 1, 2) and propentofylline (**e**,**f**: 0.1, 1, 10, 100). All the active priciples concentrations are expressed in mM. The graphs show the mean + SEM of three independent experiments performed in triplicate. The statistically significant differences between experimental conditions and untreated control cells are shown by asterisks (* *p <* 0.01; ** *p <* 0.01; *** *p <* 0.001).

**Figure 7 ijms-23-07175-f007:**
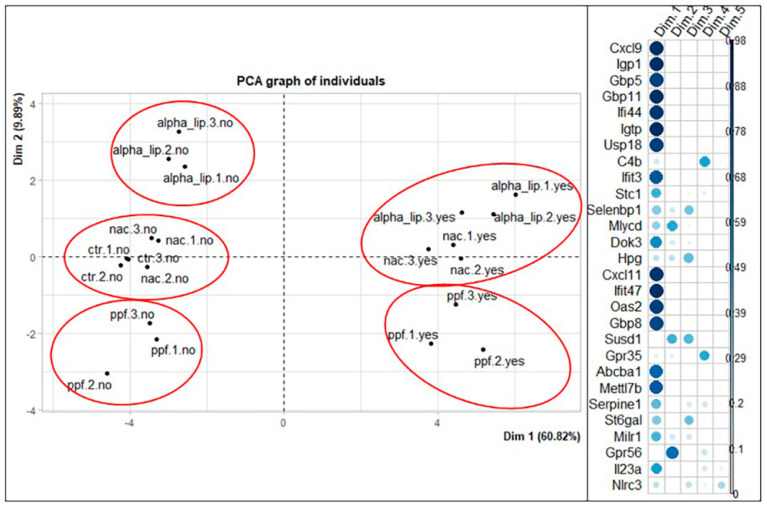
Representation of the distribution of samples in the new coordinate system obtained through the PCA technique for HT22 cell line. Correlation plot shows how much a gene contributes to each dimension of the new coordinate system. alpha_lip: α-lipoic acid; nac: N-acetyl-cystein; ppf: propentofylline; yes/no: presence/absence of active principle treatment; 1/2/3: replicates.

**Figure 8 ijms-23-07175-f008:**
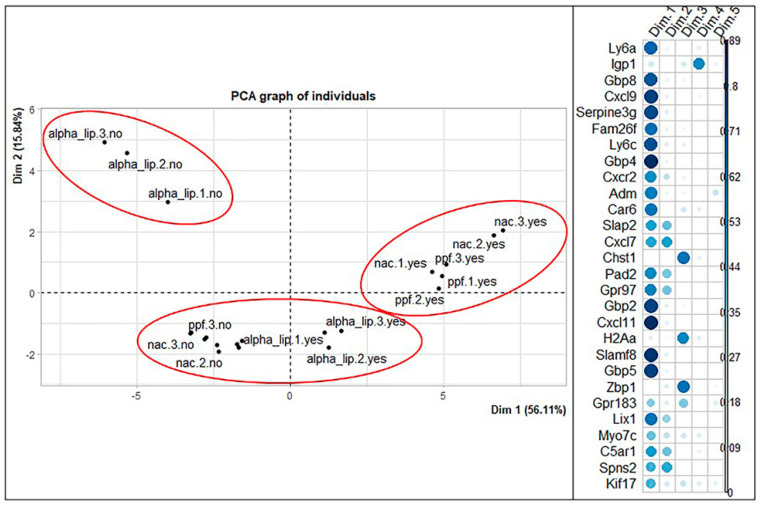
Representation of the distribution of samples in the new coordinate system obtained through the PCA technique for BV2 cell line. Correlation plot shows how much a gene contributes to each dimension of the new coordinate system. alpha_lip: α-lipoic acid; nac: N-acetyl-cystein; ppf: propentofylline; yes/no: presence/absence of active principle treatment; 1/2/3: replicates.

**Table 1 ijms-23-07175-t001:** Fold change ± SEM of inflammatory marker genes in HT22 compared with non-inflamed cells. The statistically significant differences between treated and untreated control cells are shown by asterisks (* *p* < 0.05, ** *p* < 0.01) and bold type.

** *HT22 Upregulated* **	** *Ctr* **	** *α-lipoic acid* **	** *N-acetyl-cysteine* **	** *propentofylline* **
*Cxcl9*	1 ± 0.19	0.65 ± 0.8	**0.64 ± 0.02 ***	0.78 ± 0.19
*ligp1*	1 ± 0.12	0.85 ± 0.1	**0.84 ± 0.06 ***	0.9 ± 0.16
*Gbp5*	1 ± 0.05	**0.94 ± 0.04 ***	0.79 ± 0.22	0.57 ± 0.12
*Gbp11*	1 ± 0.14	1 ± 0.04	1.05 ± 0.1	0.84 ± 0.09
*Ifi44*	1.18 ± 0.11	1.18 ± 0.05	**0.85 ± 0.12 ***	0.8 ± 0.11
*Cxcl11*	1 ± 0.09	**1.37 ± 0.17 ***	0.97 ± 0.1	0.87 ± 0.11
*Igtp*	1 ± 0.13	**1 ± 0.14 ***	0.86 ± 0.11	0.96 ± 0.04
*Ifi47*	1 ± 0.03	0.89 ± 0.1	0.96 ± 0.08	0.95 ± 0.13
*Usp18*	1 ± 0.1	1.16 ± 0.02	1.11 ± 0.22	1.55 ± 0.13
*Ifit3*	1 ± 0.08	1.41 ± 0.18	0.65 ± 0.08	1.43 ± 0.09
*Oas2*	1 ± 0.13	0.93 ± 0.09	**0.59 ± 0.05 ***	0.72 ± 0.11
*Gbp8*	1 ± 0.12	1.12 ± 0.06	1.24 ± 0.19	0.75 ± 0.12
*C4b*	1 ± 0.1	**0.67 ± 0.1 ***	**0.71 ± 0.05 ***	1 ± 0.05
** *HT22 downregulated* **	** *ctr* **	** *α-lipoic acid* **	** *N-acetyl-cysteine* **	** *propentofylline* **
*Stc1*	1 ± 0.06	0.76 ± 0.11	**0.46 ± 0.08 ****	0.5 ± 0.08 *
*Susd1*	1 ± 0.15	1.35 ± 0.06	0.54 ± 0.14	0.58 ± 0.18
*Selenbp1*	1 ± 0.03	0.52 ± 0.13	1.16 ± 0.16	0.54 ± 0.12
*Nlrc3*	1 ± 0.06	0.83 ± 0.23	**0.88 ± 0.13 ***	0.84 ± 0.02
*Gpr35*	1 ± 0.16	1.06 ± 0.02	1.12 ± 0.03	0.78 ± 0.07
*Mlycd*	1 ± 0.16	0.96 ± 0.13	0.98 ± 0.1	0.62 ± 0.02
*Dok3*	1 ± 0.07	1 ± 0.23	0.89 ± 0.12	1.09 ± 0.25
*Abcb1a*	1 ± 0.09	0.99 ± 0.05	1 ± 0.05	1 ± 0.15
*St6gal1*	1 ± 0.05	0.69 ± 0.05	0.99 ± 0.13	0.85 ± 0.14
*Milr1*	1 ± 0.07	**2.95 ± 0.26 ***	1.42 ± 0.1	0.94 ± 0.07
*Hpg*	1 ± 0.05	**3.5 ± 0.88 ***	0.8 ± 0.27	**1.2 ± 0.15 ***
*Gpr56*	1 ± 0.14	**1.37 ± 0.04 ***	1.04 ± 0.09	0.87 ± 0.17
*Il23a*	1 ± 0.03	1.01 ± 0.13	0.99 ± 0.09	0.87 ± 0.17
*Mettl7b*	1 ± 0.04	1.1 ± 0.14	0.99 ± 0.12	1.1 ± 0.05
*Serpine1*	1 ± 0.01	2.53 ± 0.2	1.11 ± 0.09	1.11 ± 0.18

**Table 2 ijms-23-07175-t002:** Fold change ± SEM of inflammatory marker genes in BV2 compared with non-inflamed cells. The statistically significant differences between treated and untreated control cells are shown by asterisks (* *p <* 0.05, ** *p <* 0.01) and bold type.

** *BV2 Upregulated* **	** *Ctr* **	** *α-lipoic acid* **	** *N-acetyl-cysteine* **	** *propentofylline* **
*Ly6a*	1 ± 0.02	0.22 ± 0.05	**0.02 ± 0.01 ***	1.16 ± 0.09
*Iigp1*	1 ± 0.02	0.09 ± 0.03	**0.04 ± 0 ***	1.47 ± 0.12
*Gbp8*	1 ± 0.1	**0.57 ± 0 ***	1.66 ± 0.02	0.84 ± 0.12
*Cxcl9*	1 ± 0.07	0.14 ± 0.03	1 ± 0.22	0.93 ± 0.28
*Serpine3G*	1 ± 0.13	0.18 ± 0.03	0.87 ± 0.05	0.58 ± 0.08
*Fam26f*	1 ± 0	**0.11 ± 0 ***	0.83 ± 0.02	0.44 ± 0.02
*Ly6c1*	1 ± 0.07	**0.16 ± 0.02 ***	1.21 ± 0.01	0.88 ± 0.01
*Gbp2*	1 ± 0.07	0.34 ± 0.11	1.36 ± 0.35	1.06 ± 0.05
*Cxcl11*	1 ± 0.08	0.32 ± 0.02	1.34 ± 0.02	0.83 ± 0.02
*Gbp4*	1 ± 0.03	0.33 ± 0.06	1.08 ± 0.04	0.97 ± 0.09
*H2-Aa*	1 ± 0.04	0.1 ± 0	2.46 ± 0.02	1.43 ± 0.07
*Slamf8*	1 ± 0.1	0.12 ± 0.04	1.26 ± 0.05	1 ± 0.18
*Gbp5*	1 ± 0.04	1.16 ± 0.53	**1.36 ± 0.11 ****	1.45 ± 0.18
*Zbp1*	1 ± 0.1	0.8 ± 0.07	**0.49 ± 0.05 ***	0.62 ± 0.01
** *BV2 downregulated* **	** *ctr* **	** *α-lipoic acid* **	** *N-acetyl-cysteine* **	** *propentofylline* **
*Cxcr2*	1 ± 0.02	1.41 ± 0.09	1.12 ± 0.01	1.25 ± 0.11
*Adm*	1 ± 0.05	1.36 ± 0.08	0.91 ± 0.06	0.82 ± 0.03
*Lix1*	1 ± 0.08	1.52 ± 0.09	**1.99 ± 0.35 ****	1.52 ± 0.14
*Gpr183*	1 ± 0.04	0.07 ± 0.04	1.58 ± 0.16	0.25 ± 0.03
*Myo7b*	1.01 ± 0.17	3.12 ± 0.55	1.2 ± 0.06	1.31 ± 0.01
*Btc*	1 ± 0.07	1.34 ± 0.03	0.97 ± 0.07	1.16 ± 0.03
*C5ar1*	1 ± 0.07	1.29 ± 0.09	1 ± 0.02	1.55 ± 0.1
*Spns2*	1 ± 0.07	0.49 ± 0.05	1.77 ± 0.41	0.63 ± 0.03
*Car6*	1 ± 0.07	0.93 ± 0.02	1.09 ± 0.09	0.88 ± 0.01
*Kif17*	1 ± 0.1	**4.38 ± 0.29 ****	3.03 ± 0.82	**3.91 ± 0.42 ***
*Sla2*	1 ± 0.08	**1.4 ± 0.06 ****	1.29 ± 0.07	1.86 ± 0.11
*Cxcl7*	1 ± 0.06	1.39 ± 0.08	1.09 ± 0.1	1.16 ± 0.05
*Chst1*	1 ± 0.02	1.09 ± 0.11	0.77 ± 0.1	0.96 ± 0.11
*Padi2*	1 ± 0.08	3.07 ± 0.14	1.06 ± 0.08	1.04 ± 0.15
*Gpr97*	1 ± 0	1.12 ± 0.03	0.76 ± 0.03	1.02 ± 0.01

**Table 3 ijms-23-07175-t003:** Set of dysregulated target genes. Selection criteria: involvement in neurodegenerative disease progression in mouse model (in black) and human (in red), as reported in references, and high rate of prediction in our in vitro model highlighted by PCA analysis of treatment-affected genes.

**HT22**	**References**
*Cxcl9*	[19,20,21]
*Iigp1*	[22,23]
*Gbp5*	[24]
*Ifi44*	[25,26]
*C4b*	[27]
*Stc1*	[28,29]
*Nlrc3*	[30]
*Abcb1a*	[31,32]
*Hp*	[33]
**BV2**	**References**
*Ly6a*	[34]
*Cxcl9*	[19,20,21]
*Serpine3G*	[35,36]
*Fam26f*	[37]
*Gbp2*	[38]
*Slamf8*	[24]
*Gbp5*	[24]
*Adm*	[39,40]

## Data Availability

Not applicable.

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
