# Peer review of "A Set of Dysregulated Target Genes to Reduce Neuroinflammation at Molecular Level"

_ijms, 2022, doi:10.3390/ijms23137175_

Round 1
Reviewer 1 Report
The work described in this manuscript was aimed at providing an in vitro model (based on co-cultures of mouse hippocampal HT22 cells and inflamed microglia BV2 cells) to screen neuroprotective agents, by selecting a group of inflammatory marker genes linked to neurodegenerative disorders using next generation sequencing.
Although the aims are clear, the results should be revised to improve their clarity for general readers.
1) Although the Methods' section all used protocols are described, the Results are overly summarized.
I feel a short "introduction" would help the reader by providing the data in the context of each subsection.
The Authors should try to insert few sentences to contextualize the results in each Results' subsection without the need for the reader to go to the M&Methods section to understand the experiment.
For example: in the Material and Methods, it is written "...bioactive molecules α-lipoic acid, N-acetyl-cystein and propentophylline, already known for their neuronal activity."
For general readers, it should be helpful to explain (even very shortly) the putative effects of these compounds in the pertinent section of the Results, and to add some supporting references (beyond those already cited, line 71).
2) 2.2 Effect of stimulated BV2 cells on differentiated HT22
It is better to specify type of effect in the title. For example: "Effect of stimulated BV2 cells on the viability of differentiated HT22 "
3) Table S2. Effect of inflammation and co-culture factors and their interaction.
What does it mean "F (1,8)"?
4) Page 5, line 106.
It is written: "Modulation of DEG identified through mRNA sequencing, shown in Table S1, was confirmed by repeating the co-culture experiment and comparing gene expression with non-inflamed cells cultured in standard monolayer. More specifically, with the aim of understanding the effect of co-culture on gene expression, both HT22 and BV2 cultured in standard monolayer were stimulated with LPS 100 ng/ml and IFNγ 5 ng/ml for 24 hours. Table S2 shows the effect of the “inflammation” and “co-culture” factors and the interaction between them."
a) For a general reader, it is not clear what does the "interaction" means. It should be explained in the main text or, at least, in the legend to the Table.
b) It is ok that this experiment highlights the direct effects of LPS and IFNγ on the single cell lines, "bypassing" the effect of co-cultures.
However, it is not clear how this information was used, above all considering the data presented in Figure 3.
The resolution of the figure is low, but it seems that the number of genes (circles) on the left side (corresponding to the downregulated/upregulated proteins in BV2 and HT22 cells) is different from those presented in Table S2 and S3 (or even higher).
5) The section "4.4 Pellet, mRNA isolation and gene expression" is duplicated.
Reviewer 2 Report
This is an interesting study which assesses the viability of an in vitro differentiated HT22 cell model to screen neuroprotective agents by the use of inflammatory marker genes.
I have a few comments for the authors.
1. What do the authors mean by the `top twenty differentially expressed genes` that were selected as inflammatory markers? I was surprised that NRF2 or other antioxidant genes were not selected.
2. What was the rationale for bioactive compounds used in this study and are they known to cross the blood brain barrier and be taken up by neuronal cells?
3. Can differentiated HT22 cells be considered an appropriate model for human neuronal cells ? What is the rationale?
4. Some information that the products of the genes selected in this study would be appropriate, possibly in a table. Is there any putative mechanism the authors can provide for why expression of these genes is affected by the neuroinflammatory response?
Round 2
Reviewer 1 Report
The authors addressed all of my concerns, and I consider the revised manuscript suitable for publication.